# Regulation of Inflammation-Mediated Endothelial to Mesenchymal Transition with Echinochrome a for Improving Myocardial Dysfunction

**DOI:** 10.3390/md20120756

**Published:** 2022-11-30

**Authors:** Byeong-Wook Song, Sejin Kim, Ran Kim, Seongtae Jeong, Hanbyeol Moon, Hojin Kim, Elena A. Vasileva, Natalia P. Mishchenko, Sergey A. Fedoreyev, Valentin A. Stonik, Min Young Lee, Jongmin Kim, Hyoung Kyu Kim, Jin Han, Woochul Chang

**Affiliations:** 1Institute for Bio-Medical Convergence, Catholic Kwandong University International St. Mary’s Hospital, Incheon 22711, Republic of Korea; 2Department of Biology Education, College of Education, Pusan National University, Busan 46241, Republic of Korea; 3G.B. Elyakov Pacific Institute of Bioorganic Chemistry, Far-Eastern Branch of the Russian Academy of Science, 690022 Vladivostok, Russia; 4Department of Molecular Physiology, College of Pharmacy, Kyungpook National University, Daegu 41566, Republic of Korea; 5Department of Life Systems, Sookmyung Women’s University, Seoul 04310, Republic of Korea; 6Department of Physiology, College of Medicine, Cardiovascular and Metabolic Disease Center, Smart Marine Therapeutic Center, Inje University, Busan 47392, Republic of Korea

**Keywords:** endothelial–mesenchymal transition, echinochrome A, myocardial infarction, cardiac fibrosis

## Abstract

Endothelial–mesenchymal transition (EndMT) is a process by which endothelial cells (ECs) transition into mesenchymal cells (e.g., myofibroblasts and smooth muscle cells) and induce fibrosis of cells/tissues, due to ischemic conditions in the heart. Previously, we reported that echinochrome A (EchA) derived from sea urchin shells can modulate cardiovascular disease by promoting anti-inflammatory and antioxidant activity; however, the mechanism underlying these effects was unclear. We investigated the role of EchA in the EndMT process by treating human umbilical vein ECs (HUVECs) with TGF-β2 and IL-1β, and confirmed the regulation of cell migration, inflammatory, oxidative responses and mitochondrial dysfunction. Moreover, we developed an EndMT-induced myocardial infarction (MI) model to investigate the effect of EchA in vivo. After EchA was administered once a day for a total of 3 days, the histological and functional improvement of the myocardium was investigated to confirm the control of the EndMT. We concluded that EchA negatively regulates early or inflammation-related EndMT and reduces the myofibroblast proportion and fibrosis area, meaning that it may be a potential therapy for cardiac regeneration or cardioprotection from scar formation and cardiac fibrosis due to tissue granulation. Our findings encourage the study of marine bioactive compounds for the discovery of new therapeutics for recovering ischemic cardiac injuries.

## 1. Introduction

Myocardial infarction (MI) usually results from an atherosclerotic plaque rupture and coronary artery occlusion after thrombosis [1]. Ischemia of the cardiac tissue downstream of the blocked vessel can lead to death or necrosis of myocardial cells; however, depending on the severity of the injury, it causes an inflammatory response that gradually removes the damaged area, leaving scar tissue with dilated capillaries. The activation and proliferation of endothelial cells (ECs) and infiltration of myofibroblasts are induced when abnormal cells are replaced and filled with granulation tissue [2]. In an attempt to restore the blood supply, angiogenesis is induced, leading to the formation of new blood vessels, while myofibroblasts deposit collagen and other extracellular matrix proteins to create scarring. In the regenerative process of a congenitally low heart, excessive scar formation weakens the ability of electromechanical coupling between cardiomyocytes, making it difficult to control contractile function. Therefore, the regeneration process of damaged myocardium according to angiogenesis and the process of generating cells necessary for the induction and generation of vascular ECs present in the myocardium are important.

Endothelial–mesenchymal transition (EndMT), which is closely related to epithelial–mesenchymal transition (EMT), is an important process for generating mesenchymal cells required for heart and valve development within the endocardium [3]. In this pathological mechanism, the microenvironment at the interface between interleukin-1β (IL-1β) and transforming growth factor-β2 (TGF-β2) can be strongly regulated [4]. Recent lineage tracing studies have reported that cardiovascular development and pathogenesis switch with each other, following changes in endothelial and mesenchymal cell types during the EndMT phase [3,5]. During the embryonic stage, endocardial ECs generate approximately 20% of the pericardial and smooth muscle cells via EndMT [6,7]. However, excessive EndMT induction generates myofibroblasts and causes myocardial fibrosis or interferes with EC homeostasis, causing atherosclerosis, decreased vascular function, and myocardial fibrosis after cardiac stress or injury [7,8,9,10]. These are major aspects of pathological hypertrophy, including cell death, mitochondrial dysfunction, dysregulation of calcium-handling proteins, and inadequate angiogenesis [11,12].

Therefore, there is a need for a process that can properly regulate EndMT during the initial stage of ischemic heart induction.

Echinochrome A (2,3,5,7,8-pentahydroxy-6-ethyl-1,4-naphthoquinone, EchA), a natural dark-red pigment with antioxidant and anti-inflammatory effects isolated from sea urchin shells, has been identified as a marine bioactive compound [13,14]. Furthermore, we have previously shown that EchA can selectively modulate calcium ions in cardiac excitation–contraction and induce the effect of reducing ischemia-induced ventricular repolarization changes, as well as protecting mitochondrial function in response to myocardial toxicity [15,16,17]. Interestingly, the discovery of protein kinase C-iota (PKCι) as a novel molecular target for EchA has indicated that it could help in drug repositioning for various diseases [18]. However, the mechanism by which EchA exerts a myocardial protective effect in ischemic tissues remains unclear.

In this study, we investigated the effects of EchA on ECs during EndMT. In addition, we examined the effects of EchA on EndMT during the inflammatory process of ischemic hearts. To the best of our knowledge, our study demonstrates for the first time that EchA negatively regulates EndMT and decreases the myofibroblast ratio and fibrosis area, which serve as a useful treatment for cardioprotection.

## 2. Results

### 2.1. EchA Inhibits EndMT Induced by TGF-β2 and IL-1β in HUVECs

Co-stimulation with TGF-β2 and IL-1β, but not TGFβ1, synergistically induces EndMT [4]. Moreover, EchA has demonstrated antioxidant and anti-inflammatory effects and has been applied to various inflammatory and fibrotic diseases [19]. Therefore, we examined the possibility that EchA participates in the anti-fibrotic process by inhibiting TGF-β2- and IL-1β-induced EndMT in HUVECs. Control HUVECs exhibited a cobblestone-like morphology, whereas TGF-β2- and IL-1β-stimulated HUVECs exhibited an elongated spindle-shaped morphology, which is a characteristic feature of mesenchymal cells (Figure 1A). Changes in EndMT under ischemic conditions were assessed using an MI model. After sacrifice, changes in TGF-β2 and IL-1β expression levels were confirmed. Compared to normal tissue, IL-1β and TGF-β2 levels in ischemic heart tissue were elevated by 12.2-fold and 2.6-fold, respectively (Figure 1C).

To confirm whether these changes can be regulated at the cellular level, such as changes in EndMT factors in tissues, a dose-dependent change in EchA was observed. No significant effect on cell viability was observed after treatment with EchA (0–10 μM) for 24 h (Figure 1D). Therefore, after cells were pretreated with EchA for 24 h, they were co-stimulated with TGF-β2 and IL-1β for 5 days. We then evaluated the expression of endothelial cell markers CD31 and VE-cadherin and mesenchymal cell markers α-SMA and fibronectin by Western blotting. The results demonstrated that EchA significantly reversed the TGF-β2- and IL-1β-stimulated increase in α-SMA and fibronectin expression and TGF-β2- and IL-1β-stimulated decrease in CD31 and VE-cadherin expression in a dose-dependent manner (Figure 1E,F). To further confirm the transition of HUVECs to mesenchymal cells, HUVECs were exposed to TGF-β2 and IL-1β for five days and then labeled with FL-LDL. The results revealed that cells were unable to take up LDL in the presence of TGF-β2 and IL-1β, demonstrating that the HUVECs lost their endothelial properties and underwent a transition. Moreover, the treatment of HUVECs with EchA preserved LDL uptake in a dose-dependent manner, thereby preventing cell transformation (Figure 1G,H). A transwell assay was performed to analyze the intercellular permeability of FITC-labeled dextran. The permeability of HUVECs was significantly increased after co-stimulation with TGF-β2 and IL-1β, compared with the negative control group. However, the treatment of these cells with EchA significantly reversed this effect (Figure 1I). These results suggest that EndMT events occur in an ischemic heart, and EndMT-associated TGF-β2- and IL-1β-stimulated HUVECs are regulated by EchA in a dose-dependent manner at the cellular level.

### 2.2. EchA Suppresses Cell Migration and Rho GTPase Pathway Activation

During EndMT, cell migration is promoted [20]. In the wound healing assay, EchA suppressed the migration of TGF-β2- and IL-1β-stimulated HUVECs (Figure 2A,B). Further evaluation by transwell assay demonstrated increased cell migration in the TGF-β2- and IL-1β-stimulated groups compared with the control group, and EchA significantly suppressed this effect (Figure 2C,D).

Non-Smad pathways are activated during EndMT, including small GTPases, such as RhoA, which is involved in actin and microtubule cytoskeleton organization [21]. Compared with the control group, RhoA levels and Cdc42 and cofilin phosphorylation were significantly increased in the TGF-β2- and IL-1β-stimulated groups. In contrast, treatment with 10 μM EchA prevented the TGF-β2- and IL-1β-induced phosphorylation of Cdc42 and cofilin and expression of RhoA (Figure 2E,F). These results indicate that EchA inhibits cell migration during EndMT via a non-Smad pathway.

### 2.3. EchA Downregulates NF-κB and Smad Pathway and Reduces the Intracellular ROS Level

It has been reported that NF-κB is essential for both inducing and maintaining EndMT [22]. In addition, inflammatory signals and increased ROS production facilitate EndMT by increasing endogenous TGF-β expression in an NFκB-dependent manner, creating a feed-forward signaling mechanism [23]. Therefore, we conducted an experiment to determine the effects of EchA on ROS formation induced by TGF-β2 and IL-1β stimulation in HUVECs. Compared with the control group, ROS formation was significantly increased in the TGF-β2- and IL-1β-stimulated group; however, the treatment of the TGF-β2- and IL-1β-stimulated group with EchA decreased ROS production to nearly control levels (Figure 3A,B). Smad2/3 and NF-κB p65 phosphorylation was upregulated in the TGF-β2- and IL-1β-stimulated group compared with the control. However, this effect was also inhibited by EchA (1, 3, 5, and 10 μM) treatment of TGF-β2- and IL-1β-stimulated cells (Figure 3C,D). These results suggest that EchA can regulate inflammatory signals and ROS levels in ECs undergoing EndMT in a dose-dependent manner.

### 2.4. EchA Improves Mitochondrial Dysfunction of TGF-β2- and IL-1β-Induced EndMT in HUVECs

Mitochondrial dysfunction can induce fibrotic diseases [24]. Moreover, many studies that have investigated mitochondrial function during EMT have found that TGF-1β treatment impairs mitochondrial function by increasing ROS production and reducing mitochondrial membrane potential, ATP content, mitochondrial DNA content, and mitochondrial complex protein expression [25,26,27]. Our recent study demonstrated that EchA significantly increased mitochondrial mass and oxidative phosphorylation function, which enhanced the mitochondrial energy efficiency by modulating major mitochondrial biogenesis regulatory genes, including PGC-1α and NRF-1 [28]. Therefore, we examined whether EchA improves mitochondrial dysfunction by inhibiting TGF-β2- and IL-1β-induced EndMT in HUVECs. First, the mitochondrial membrane potential was measured using TMRE fluorescence. Compared with the control group, the mitochondrial membrane potential was significantly decreased in the TGF-β2- and IL-1β-stimulated group. Conversely, EchA (1, 3, 5, and 10 μM) treatment significantly increased the mitochondrial membrane potential of the TGF-β2- and IL-1β-stimulated group (Figure 4A,B). In addition, compared with the TGF-β2- and IL-1β-stimulated group, mitochondrial DNA content significantly increased in the EchA (10 μM) treatment group (Figure 4C,D). To evaluate the mitochondrial function, we examined the expression levels of mitochondrial biogenesis-regulated genes using qPCR. The results of qPCR demonstrated that the expression levels of these genes were significantly increased after treatment with EchA, compared with the TGF-β2- and IL-1β-stimulated groups (Figure 4E). Overall, these results suggest that ECs undergoing EndMT have a protective effect against mitochondrial dysfunction, due to EchA treatment.

### 2.5. EchA Positively Regulates the EndMT Process in Ischemic Hearts

EndMT refers to the process through which ECs transition to a mesenchymal-like cell state, including myofibroblasts or smooth muscle cells; however, there are no agreed-upon criteria for defining this at the molecular level [29]. Therefore, we conducted a study using an induced MI model, a condition that can express mesenchymal genes, such as SM22α [30,31,32]. Based on the results of our previous study, EchA was used at a concentration of 0.1 mg/kg body weight for this experiment. EchA was intraperitoneally injected in mice on days 0, 1, and 2 after MI induction (Figure 5A).

To verify the effect of EchA on the EndMT process, EndMT-related genes were measured in mouse cardiac tissue from MI or EchA-treated MI. EndMT genes, including *Col1a*, *Fsp1, Snail, Tgfb*, and vimentin, are known to be altered in ischemic conditions [33,34,35,36,37]. We also examined these changes (Figure 5B). Compared to the control group, the expression of *Col1a*, *Fsp1*, *Snail*, *Tgfb*, and *Vimentin* increased 9.2-, 3.6-, 1.7-, 2.3-, and 2.5-fold, respectively, in the MI group. In the EchA-treated MI group, the expression levels of these genes decreased to almost normal levels. Double immunofluorescence staining, performed to detect the expression of the endothelial junction protein VE-cadherin and mesenchymal marker α-SMA, was consistent with the evidence for EndMT in the MI group (Figure 5C). In contrast, cells that express mesenchymal markers were rarely detected in the EchA-treated MI group. MMP-2 and MMP-9 are involved in myocardial remodeling after MI, and collagen I and III, well-known fibrosis-related factors [30,38], were identified at the protein level. We confirmed that the relatively high protein expression rate in MI was downregulated after EchA treatment (Figure 5D,E). Given this finding, it is important to demonstrate that EndMT is effectively regulated by EchA treatment in response to ischemic conditions in the heart.

### 2.6. EchA Exerts Cardioprotective Effects by Preventing EndMT in Ischemic Hearts

The treatment efficiency according to the in vivo regulation of EndMT in MI was evaluated. Histological changes were observed at week 1, when EchA was absorbed into the body and granulation tissue formation was determined. Functional analysis of the myocardium was performed at week 4 when scar formation matured. By measuring infarct size using TTC staining, it was confirmed that the ischemic area was significantly reduced in the EchA-treated MI group, compared with the MI group (Figure 6A). EchA treatment also significantly attenuated cardiac apoptosis, as assessed by the TUNEL assay (Figure 6B). In addition, due to the inhibition of EndMT, the area of fibrosis was significantly decreased in the EchA-treated MI group, compared to that in the MI group (Figure 6C). To determine the phenotype ratio of ECs that did not transition to the mesenchymal phenotype, the degree of angiogenesis was investigated using the CD31 antibody. In the EchA-treated MI group, it was possible to confirm microvessel expression by more than 2.5-fold (Figure 6D). Millar catheterization was used to measure cardiac function based on the pressure–volume relationship. Compared with the MI group, the EchA-treated MI group showed improved load-independent parameters, such as end-systolic elasticity and the end-systolic pressure–volume relationship, as well as load-dependent parameters, including the ejection fraction and end-systolic volume (Figure 6E–G). Taken together, these results suggest the possibility of using EchA as a potential EndMT control tool for cardiac repair or cardioprotection.

## 3. Discussion

In this study, we demonstrated the role of EchA in the EndMT process of ECs and its beneficial effects in TGF-β2- and IL-1β-stimulated ECs and inflammation-related EndMT in an ischemic heart model. We observed that EndMT occurred in the ischemic heart, and the activity of TGF-β2 and IL-1β in HUVECs induced EndMT in a dose-dependent manner. We also showed that EchA reduces cell migration via the non-Smad signaling pathway in ECs and that it normally regulates inflammatory and oxidative responses and mitochondrial dysfunction. When the EndMT process based on the ischemic condition of the heart was induced, in vivo treatment with EchA prevented ECs from reaching the mesenchymal phenotype. Thus, histological and functional improvements of the myocardium were studied. This study is the first to identify the role of EchA, a marine biological activator, as a regulator of EndMT in heart diseases, including MI and fibrosis.

After an acute myocardial injury, the cardiac tissue undergoes a three-step time course (inflammation, proliferation, and remodeling/scar formation) after the removal of dead cells to heal the infarct scar and return it to its full functional capacity as much as possible [2]. To regulate this timing, local fibroblasts migrate to the injury site and are converted into myofibroblasts. Several studies have shown that they are related to the recruitment of progenitor/stem cells that are capable of transdifferentiation, which implies the EMT process [39,40,41,42]. From another point of view, the fate of the scar matters. As cardiomyocytes are generated at the final stage of differentiation, cell cycle proliferation is difficult. Therefore, they may not regenerate sufficiently and myofibroblasts may persist for many years. During the initial healing process, it is important to control the transition of ECs to myocardial fibroblasts, which is called EndMT, in cardiac tissue. Numerous mouse lineage tracing studies have demonstrated that EndMT is a source of fibroblasts and myofibroblasts and that EndMT controls fibrosis in kidney, lung, brain, and heart disease [8,43,44,45,46,47,48]. In particular, TGF-β, which is produced in a chronic inflammatory environment, is known to be a potent inducer of EndMT, and its effect on EC is being studied. We confirmed that EndMT can occur at the cellular level by treating ECs with TGF-β2 and inflammatory IL-1β, and at the same time, the potent factor in ischemic myocardial tissue rapidly increased by 2.6-fold and 12.2-fold, respectively (Figure 1A,C). Furthermore, when ECs are exposed to EndMT, intercellular adhesion is attenuated due to the decreased expression of endothelial-specific cell adhesion markers, such as VE-cadherin. In contrast, when the expression of mesenchymal cell markers is increased, cells acquire mesenchymal properties, such as an enhanced migratory ability [49]. We confirmed that wound recovery and cell migration rates decreased in a concentration-dependent manner when ECs were treated with EchA, which was expected to negatively modulate EndMT (Figure 1E,F). In addition, although co-expression of VE-cadherin and α-SMA was confirmed in ischemic heart tissue, their expression was hardly confirmed in EchA-treated tissues (Figure 5C). This means that, unlike the MI group, which has both mesenchymal and endothelial markers and has many changed areas, EchA shows a positive effect without phenotypic changes. Based on these results, it can be inferred that EchA can selectively control mesenchymal transition by regulating EndMT initiation.

EchA, a marine bioactive substance, was proposed by Boguslavskaya et al. as a new natural antioxidant and a quinonoid pigment in sea urchins. In addition, various studies, including those on ROS removal, interaction of lipid peroxide radicals, and cell redox regulation, have been reported to reveal the mechanisms of antioxidant properties [50,51,52]. EchA has unique physicochemical properties; as a histochrome, it has been shown to have therapeutic potential for eye diseases, cardiovascular diseases, and inflammatory and metabolic diseases [19]. We confirmed that the treatment of TGF-β2- and IL-1β-stimulated ECs with EchA decreased the intracellular ROS levels in a dose-dependent manner (Figure 3A,B). We also showed that EchA treatment improved mitochondrial dysfunction (Figure 4A–E).

Functional and phenotypic cellular changes during EndMT are key factors that determine whether an endothelial phenotype is maintained or if a transition to myofibroblasts is achieved [29]. We analyzed LDL uptake at the cellular level after EchA treatment to confirm the EC characteristics of LDL cholesterol uptake (Figure 1G,H). In addition, it was confirmed that collagen production and matrix metalloproteinase-secreted proteins were decreased at the cell and tissue levels; thus, EchA was found to properly control the induction of mesenchymal characteristics (Figure 5B,D). The changes in myocardial fibrosis, which is a pathological tissue repair phenomenon caused by the accumulation of extracellular matrices secreted by myofibroblasts, confirmed that EchA treatment effectively controlled myofibroblasts and fibrotic cells by inhibiting EndMT at the in vivo level, through cell death and ischemic area and functional decline induced by myocardial fibrosis (Figure 6).

TGF-β induction of Rho signaling in epithelial cells is known to be involved in mesenchymal transition by rapidly activating various Rho GTPases, including RhoA, RhoB, Rac, and Cdc42, and is mediated by the p38 MAP kinase and phosphatidylinositol 3-kinase pathways [21,53]. We treated cells with EchA to regulate the EndMT process following stimulation with TGF-β and IL-1β and found that Rho GTPase signaling was regulated (Figure 2E,F). In addition, we previously reported that EchA has a decisive effect on the inhibition of a novel atypical PKCι [18]. According to several studies on cancer cells, it was concluded that PKC not only activates NF-κB by inhibiting IκBα in the cytoplasm to prevent translocation into the nucleus, but also regulates Rho A signaling to stimulate EMT [54,55,56,57]. Furthermore, it has been reported that atypical PKC must be regulated via Rac-JNK signaling or GSK-3beta/Snail signaling for EMT of other types of cancer [55,58,59,60]. Combining these results, it is predicted that EchA can promote overall regulation in the EndMT process by IL-1β or PKCι signaling, as well as Snail-based fibrosis by TGF-β (Figure 6H).

The limitations of this study were as follows: (1) it was impossible to confirm the same EndMT situation for each mouse and (2) it was difficult to measure the distribution of EchA in vivo.

EndMT can regulate angiogenesis- and fibrosis-induced signaling, and the EchA treatment results suggest that these two important cardiac tissue repair mechanisms are intrinsically linked. In particular, we discovered a novel therapeutic purpose of EchA; it can directly control the inflammatory response, oxidative stress, and mitochondrial dysfunction, promote angiogenesis, and modulate scar size by controlling early EndMT or inflammation-related EndMT, thereby maximizing cardiac recovery after ischemic injury. EchA is expected to contribute to the development of new therapeutics for cardiac regeneration or protection from scar formation and cardiac fibrosis due to tissue granulation.

## 4. Materials and Methods

### 4.1. Cell Culture

Human umbilical vein ECs (HUVECs; ATTC #PCS-100-010) were cultured in Endothelial Cell Growth Medium-2 BulletKitTM (EGM-2; Lonza, Walkersville, MC, USA), which contained 100 U/mL of penicillin and streptomycin, and were maintained in an incubator at 37 °C with 5% CO2. Cells between passages five and ten were used in the experiments.

### 4.2. Chemicals

EchA (6-ethyl-2,3,5,7,8-pentahydroxynaphthalene-1,4-dione) was isolated from sand dollars (Scaphechinus mirabilis) using a previously described extraction method [61]. The purity of EchA (>99%) was confirmed by liquid chromatography–mass spectrometry (Shimadzu LCMS-2020, Kyoto, Japan). We used 0.02% EchA saline solution for all experiments.

### 4.3. EndMT Induction

HUVECs were seeded in 60 mm plates at a density of 1 × 10^5^ cells and were incubated for 16–24 h. Cells were co-stimulated with transforming growth factor (TGF)-β2 (10 ng/mL) and interleukin (IL)-1β (1 ng/mL) daily for 24 h at 37 °C; media were replaced every 24 h. EndMT was confirmed to be induced on day 5.

### 4.4. Cell Viability Assay

Cell viability was evaluated using the Cell Counting Kit-8 (CCK-8; Dojindo, Kumamoto, Japan). HUVECs were seeded into 96-well plates (2 × 10^4^ cells/well) and cultured in EGM-2 medium for 16 h. The cells were treated with varying concentrations (0, 1, 3, 5, or 10 μM) of EchA for 24 h. After 3 h of incubation at 37 °C with CCK-8 reagent, the absorbance was measured at 450 nm using a microplate reader.

### 4.5. Wound Healing Assay

HUVECs were seeded at a density of 1 × 10^5^ cells/well in 12-well plates. Upon reaching 90% confluence, cells were treated with 0, 1, 3, 5, or 10 μM EchA for 24 h and were serum-starved for 12 h. After starvation, scratches were produced with 200 μL pipette tips, and the medium was replaced with a medium that contained TGF-β2 (10 ng/mL) and IL-1β (1 ng/mL). Images were captured using a microscope. The migrated area was captured at 0 and 24 h after TGF-β treatment, and the percentage of the recovered area was calculated.

### 4.6. Transwell Invasion Assay

The migration of HUVECs was assessed by performing a 24-well transwell assay (Costar Corning, Corning, NY, USA). Cells treated with 0, 1, 3, 5, or 10 μM EchA were added to the upper chamber. As a chemoattractant, 600 μL of medium supplemented with TGF-β2 (10 ng/mL); IL-1β (1 ng/mL); and 0, 1, 3, 5, or 10 μM EchA was added to the lower chamber. After incubation at 37 °C for 24 h, the cells in the upper chamber were cleaned with a cotton swab, and the migrated cells on the bottom of the membrane were fixed with methanol and stained with Giemsa. Four fields per filter were randomly selected using a light microscope.

### 4.7. Detection of Reactive Oxygen Species (ROS) Formation

The intracellular generation of ROS, as an index of oxidative stress, was determined using H2DCFDA (Ex/Em = 492–495/517–527 nm, Invitrogen, Carlsbad, CA, USA). H2DCFDA is a cell-permeable probe. After diffusing into cells, it is metabolized into non-fluorescent DCF, which is subsequently oxidized by intracellular ROS into highly fluorescent DCF. HUVECs seeded into six-well plates (2 × 10^5^ cells/well) were treated with varying concentrations of EchA, as previously described. After 1 h of incubation with TGF-β2 and IL-1β, HUVECs were rinsed with phosphate-buffered saline (PBS) and incubated with 5 µM H2DCFDA in serum-free medium in a light-free chamber for 30 min.

### 4.8. Western Blot Analysis

Cell lysates were centrifuged at 12,000 rpm for 7 min at 4 °C. Protein concentrations were determined using a Pierce bicinchoninic acid (BCA) protein assay (Thermo Fisher Scientific, Waltham, MA, USA), and 10–15 μg of protein was loaded per lane onto 10% SDS–polyacrylamide gels, and the separated proteins were transferred onto nitrocellulose membranes (Millipore, Billerica, MA, USA). The membranes were subsequently blocked with 5% bovine serum albumin (BSA) with 0.1% Tween-20 and incubated with the appropriate primary antibodies overnight at 4 °C ((fibronectin: BD Pharmigen), (CD31, VE-cadherin, β-actin, total and phospho-cofilin, total and phospho-cell division cycle (cdc) 42, Rho A, total and phospho-smad2/3: Cell Signaling, Danvers, MA, USA), (total and phospho-NF-κB p65, and total and phospho-inhibitor of NF (I)κBα: Santa Cruz Biotechnology, Dallas, TX, USA); (α-smooth muscle actin (SMA) and TGF-βR1: Abcam, Cambridge, UK)). The bands were later incubated with anti-rabbit or anti-mouse peroxidase-conjugated secondary antibodies (Thermo Fisher Scientific). The protein bands were visualized using an enhanced chemiluminescence (ECL) kit (Advansta, Menlo Park, CA, USA) and quantified using the ImageJ(1.52v, NIH)software.

### 4.9. Mitochondrial Membrane Potential Assay

To evaluate the effect of EchA on mitochondria, the mitochondrial membrane potential in cells treated with varying concentrations of EchA was compared using the TMRE-Mitochondrial Membrane Potential Assay Kit (TMRE; Ex/Em = 549/575 nm; Abcam). HUVECs seeded into six-well plates (2 × 10^4^ cells/well) were treated with varying concentrations of EchA, as previously described. After 1 h of incubation with TGF-β2 and IL-1β, the HUVECs were stained with 200 nM TMRE for 30 min at 37 °C.

### 4.10. Real Time PCR

Total RNA from HUVECs was extracted using the RiboEX reagent (Invitrogen). One microgram of total RNA was reverse-transcribed using the Revet Aid First Strand cDNA Synthesis kit (Thermo, Rockford, IL, USA), according to the manufacturer’s instructions. Real-time PCR was performed using the SYBR Premix Ex Taq (Takara, Shiga, Japan), following the manufacturer’s protocol. All reactions were performed in triplicates. The cDNA was amplified using 60 cycles of 15 s at 95 °C, 15 s at 60 °C, and 30 s at 72 °C for each gene. The expression values are presented relative to the β-actin values in the corresponding samples. The primers used in this study are shown in Appendix A.

### 4.11. Low-Density Lipoprotein (LDL) Uptake Assay

After cells were pretreated with EchA for 24 h, they were co-stimulated with TGF-β2 and IL-1β for 5 d. Next, 10 μg/mL fluorescently labeled (FL) LDL (BioVision Inc., Milpitas, CA, USA) was added for 4 h, and the cells were analyzed using a fluorescence microscope.

### 4.12. Permeability Assay

ECs were seeded in 24-well plates with 0.4 μm pore size transwell inserts (Fisher Scientific, Corning Inc., #353024) at a density of 1 × 10^5^ cells per insert and grown to 80–90% confluence. Next, 200 μL of EBM-2 that contained TGF-β2, IL-1β, and EchA was added to the upper chamber and 500 μL of EBM-2 was added to the lower chamber for 24 h at 37 °C and 5% CO2. Then, 200 μL of EBM-2 that contained 1 μg/mL fluorescein-5-isothiocyanate (FITC)-dextran was added to the upper chamber, and 500 μL of EBM-2 was added to the lower chamber. The 24-well plate with transwell inserts was incubated for 1 h at 37 °C in 5% CO2 with slight shaking. The concentration of FITC-dextran transferred to the lower chamber was determined using a microplate reader, with excitation and emission wavelengths of 485 and 535 nm, respectively.

### 4.13. MI model and EchA Treatment

The surgical procedures were approved by the Institutional Animal Care and Use Committee of Catholic Kwandong University College of Medicine (No. CKU 01-2020-009) and the Association for Assessment and Accreditation of Laboratory Animal Care and were performed according to the guidelines and regulations for animal care. Twelve-week-old male C57BL/6 mice were divided into the following two groups: MI + PBS (*n* = 15) and MI + EchA (*n* = 15). The mice were anesthetized via intraperitoneal injection of tiletamine/zolazepam (Zoletile, 30 mg/kg body weight) and xylazine (10 mg/kg body weight), ventilated with a volume-regulated respirator (VentElite 55-7040, Harvard Apparatus, Holliston, MA, USA), and then subjected to median sternotomy. The MI model was established by ligation of the left anterior descending artery, using a 6–0 Prolene suture (Covidien, Dublin, Ireland). After EchA treatment, the muscle and skin were sutured using a 4–0 Prolene suture. For pathological and functional analyses, the mice were sacrificed at weeks 1 and 4, respectively.

EchA was obtained from the G.B. Elyakov Pacific Institute of Bioorganic Chemistry, Vladivostok, Russia. The substance was dissolved in PBS 30 min before each exercise training session. Infarcted mice were injected intraperitoneally with 0.1 mg/kg body weight of EchA in 300 μL PBS [13]. EchA was administered at a total volume of 50 μL per animal until days 0, 1, and 2 after MI induction.

### 4.14. 2,3,5-Triphenyltetrazolium Chloride (TTC) Stain

To measure the myocardial infarct area, isolated hearts were perfused with 1% TTC (Sigma-Aldrich, St. Louis, MO, USA) for 1 h at 37 °C and incubated in 4% formaldehyde overnight at 4°C. The heart sections were photographed using a digital camera (DIMIS M model, Anyang, Korea). The infarcted area was measured using ImageJ software.

### 4.15. Histology and Immunofluorescence

Five micrometer-thick longitudinal sections were cut from the apex to the base and mounted on a glass slide. Masson’s trichrome staining was performed according to standard protocols, and the fibrotic area was measured using ImageJ software. The terminal deoxynucleotidyl transferase dUTP nick end labeling TUNEL Assay Kit—BrdU-Red (Abcam, #ab66110) was used to determine the number of apoptotic and necrotic cells in the infarcted myocardium, as per the manufacturer’s instructions. TUNEL assay images were blindly captured and counted using a virtual microscope (BX51 Dot Slide; Olympus, Tokyo, Japan). To assess the relationship between the transplanted EV and ischemic myocardium, immunohistochemistry was performed. Sections were deparaffinized and incubated in 1% H_2_O_2_ to quench endogenous peroxidase. Cardiac tissues were blocked for 1 h with a mixture of 1% (*w*/*v*) BSA and 5% (*v*/*v*) horse serum and incubated with VE cadherin antibody (Abcam, UK, #ab205336) or α-SMA (Santa Cruz Biotechnology, #sc53015) at a ratio of 1:200. After washing thrice with PBS, the sections were incubated with secondary antibodies (Alexa Fluor 488, #ab150077; Alexa Fluor 594, #ab150116; at a ratio of 1:250, Abcam) for 1 h at room temperature, mounted with DAPI, and viewed under a confocal microscope (LSM 700, Carl Zeiss, Oberkochen, Germany).

### 4.16. Cardiac Catheterization

To measure invasive hemodynamics, left ventricular (LV) catheterization was performed 4 weeks after MI. A mouse pressure–volume loop catheter (SPR-839, Millar Instruments, Houston, TX, USA, #SPR-839NR) was introduced into the LV via the right carotid artery (closed-chest surgery) under anesthesia. Ventricular pressure and real-time volume loops were recorded, and all the data were analyzed using LabChart v8.1.5 software (Millar, Houston, TX, USA).

### 4.17. Statistical Analysis

Data are expressed as mean ± standard error of the mean of at least three independent experiments. Comparisons between more than two groups were performed using a one-way analysis of variance with Bonferroni correction. *p* values < 0.05 were considered as statistically significant.

## Figures and Tables

**Figure 1 marinedrugs-20-00756-f001:**
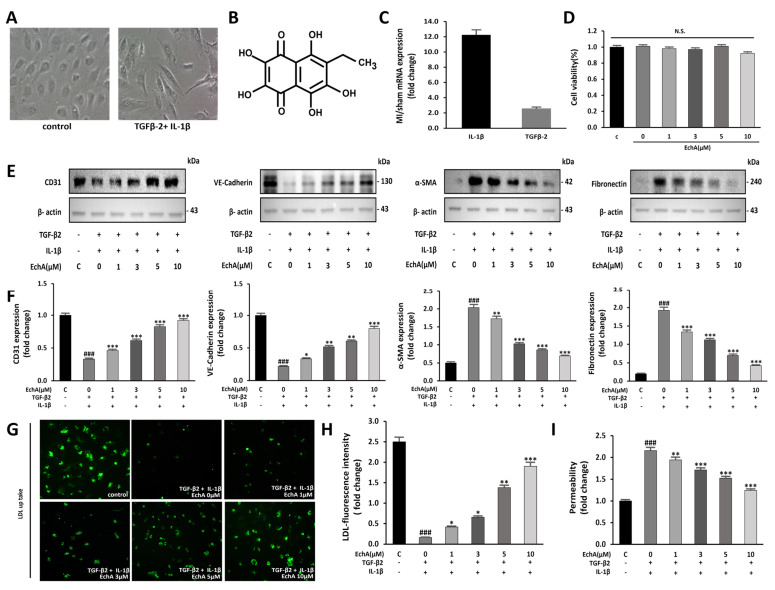
EchA inhibits EndMT in HUVECs. (**A**) EndMT is induced by TGF-β2 and IL-1β in HUVECs. Untreated HUVECs or TGF-β2 and IL-1β-treated HUVECs were examined for morphological changes using a light microscope (100 × magnification). (**B**) The chemical structure of EchA. (**C**) TGF-β2 and IL-1β mRNA expression levels were quantified in the sham control and MI groups. Compared with the sham control group, TGF-β2 and IL-1β mRNA expression levels in the MI group were increased. (**D**) The cells were pretreated with different concentrations of EchA (0, 1, 3, 5, and 10 μM) for 24 h. Cell viability was determined by the CCK-8 assay (*n* = 5). (**E**,**F**) The relative density of CD31, VE-cadherin and α-SMA, in addition to fibronectin protein bands, were analyzed by Western blot with β-actin as the control. (**G**) Fluorescence microscopic images of FL-LDL uptake in HUVECs incubated with LDL (40× magnification). (**H**,**I**) Permeability tests were performed after HUVECs were incubated with TGF-β2 and IL-1β and with different concentrations of EchA (0, 1, 3, 5, and 10 μM) for 24 h. EchA inhibits TGF-β2- and IL-1β-induced monolayer permeability of HUVECs. ^###^
*p* < 0.001 vs. HUVECs without EchA and TGF-β2 and IL-1β treatment; * *p* < 0.05, ** *p* < 0.01, *** *p* < 0.001 vs. TGF-β2- and IL-1β-induced HUVECs without EchA treatment. Data are expressed as X-fold induction compared to normal control. N.S., not significant.

**Figure 2 marinedrugs-20-00756-f002:**
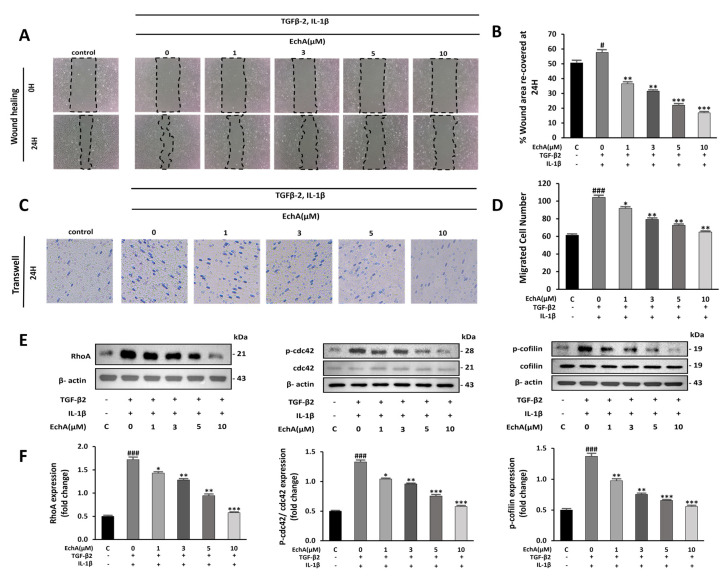
EchA suppresses migration of TGF-β2- and IL-1β-stimulated HUVECs. (**A**) Representative images of HUVECs after treatment with different EchA concentrations (0, 1, 3, 5, and 10 μM) for 24 h. Serum-starved HUVECs were first scratched using a 200 μL tip, and subsequently treated with TGF-β2 and IL-1β for 24 h. The images of the wounds were obtained at 0 and 24 h using a light microscope (40× magnification). (**B**) The area between the wound edges was measured and compared among the groups. (**C**) Representative images of migrated TGF-β2- and IL-1β-stimulated HUVECs in transwell chamber assay after treatment with different concentrations (0, 1, 3, 5, and 10 μM) of EchA for 24 h using a light microscope (40× magnification). (**D**) The number of migrated cells was measured and compared among the groups. (**E**,**F**) Relative density analysis of the RhoA protein bands by Western blot with β-actin as the control. Relative density analysis of phosphorylated Cdc42 and cofilin protein bands by Western blot with Cdc42 and cofilin as the controls. ^#^
*p* < 0.05, ^###^
*p* < 0.001 vs. HUVECs without EchA and TGF-β2 and IL-1β treatment; * *p* < 0.05, ** *p* < 0.01, *** *p* < 0.001 vs. TGF-β2- and IL-1β-induced HUVECs without EchA treatment. Data are expressed as X-fold induction compared to normal control.

**Figure 3 marinedrugs-20-00756-f003:**
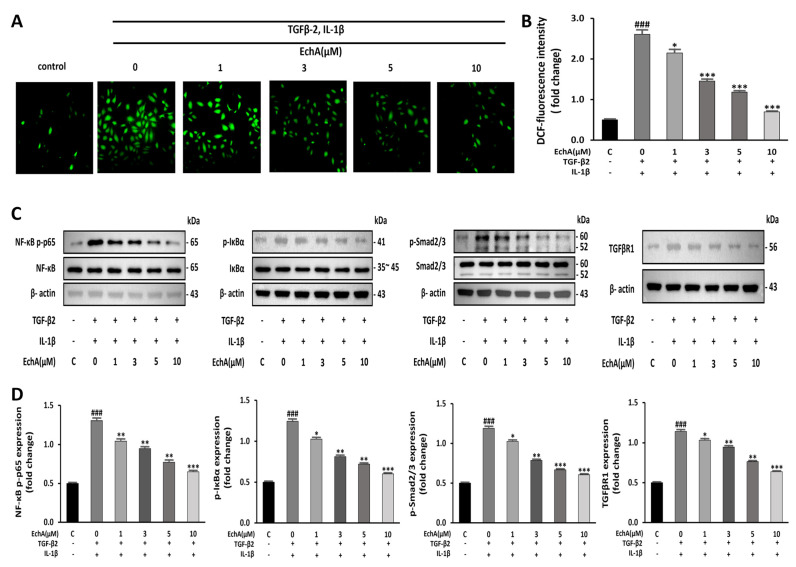
EchA induces anti-inflammatory and anti-oxidative responses in HUVECs. (**A**) ROS production in HUVECs was determined by DCFH-DA assay. The HUVECs were pretreated with different concentrations of EchA for 24 h prior to treatment with TGF-β2 and IL-1β. Images were taken using a fluorescence microscope (40× magnification). (**B**) The bar graph shows the statistical results of fluorescence intensity. The mean intensity of DCFH-DA was quantified using ImageJ software. (**C**,**D**) Relative density analysis of the NF-κB p-p65, p-IκBα, and p-smad2/3 protein bands by Western blot with NF-κB p65, IκBα, and smad2/3 as the control. Relative density analysis of the TGFβR1 protein bands by Western blot with β-actin as the control. ^###^
*p* < 0.001 vs. HUVECs without EchA and TGF-β2 and IL-1β treatment; * *p* < 0.05, ** *p* < 0.01, *** *p* < 0.001 vs. TGF-β2- and IL-1β-induced HUVECs without EchA treatment. Data are expressed as X-fold induction compared to normal control.

**Figure 4 marinedrugs-20-00756-f004:**
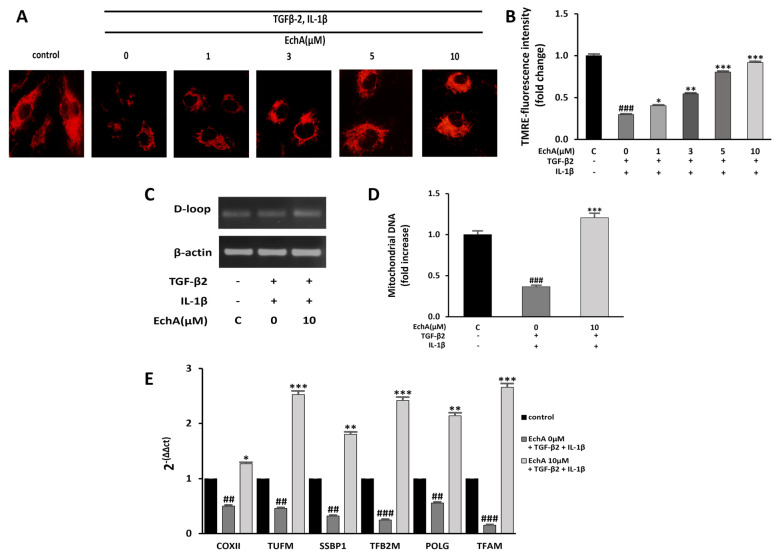
EchA improves mitochondrial dysfunction of EndMT in HUVECs. (**A**) Mitochondrial membrane potential in HUVECs was determined by TMRE assay. The HUVECs were pretreated with different concentrations of EchA for 24 h prior to treatment with TGF-β2 and IL-1β. Images were captured using a fluorescence microscope (200× magnification). (**B**) The bar graph shows the statistical results of fluorescence intensity. The mean intensity of TMRM was quantified using ImageJ software. (**C**,**D**) The mtDNA/nDNA ratio was analyzed to quantify mitochondrial DNA. (**E**) Effect of EchA on the mRNA expression levels of COXⅡ, TUFM, SSBP1, TFB2M, POLG, and TFAM, as measured by RT-qPCR and normalized to β-actin. ^##^
*p* < 0.01, ^###^
*p* < 0.001 vs. HUVECs without EchA and TGF-β2 and IL-1β treatment; * *p* < 0.05, ** *p* < 0.01, *** *p* < 0.001 vs. TGF-β2- and IL-1β-induced HUVECs without EchA treatment. Data are expressed as X-fold induction compared to normal control.

**Figure 5 marinedrugs-20-00756-f005:**
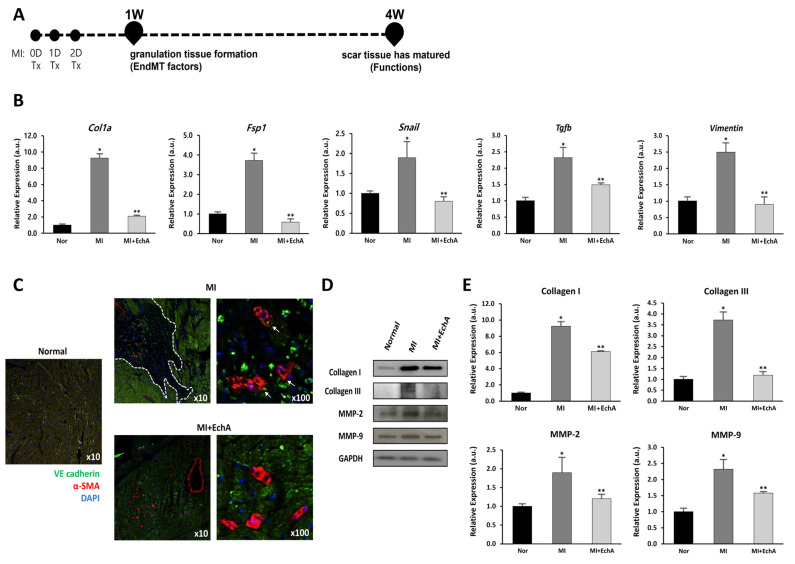
EchA regulates EndMT in the infarcted heart. (**A**) The timeline of the animal study is presented. After MI modeling, mice were administered EchA for 3 days, and histological and cardiac function analyses for EndMT factor were performed 1 and 4 weeks after surgery, respectively. (**B**) RT-qPCR was performed to analyze the expression of EMT- or EndMT-associated genes *Col1a*, *Fsp1*, *Snail*, *Tgfb*, and *Vimentin*. (**C**) Immunofluorescence image of EndMT in the infarcted area (green: VE-cadherin (endothelial origin), red: α-SMA (fibrosis), blue: DAPI, dotted white line: ischemic area, white arrow: VE-cadherin and α-SMA double positive cells) (10× and 100× magnification). (**D**,**E**) Relative density analysis of the collagen I, collagen III, MMP-2, and MMP-9 protein bands by Western blot with GAPDH as the control. * *p* < 0.05, ** *p* < 0.001 vs. normal group. Data are expressed as X-fold induction compared to normal control. All values are mean ± standard deviation. Statistical significance was assessed by one-way ANOVA.

**Figure 6 marinedrugs-20-00756-f006:**
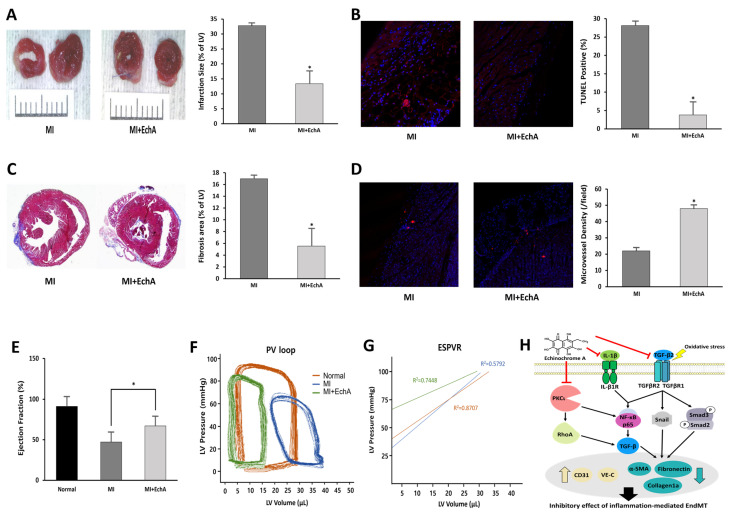
EchA exerts cardioprotective effects after myocardial infarction. (**A**) Representative photo and column scatter plot of infarct size determined through TTC staining. Infarct size (%) was calculated as the ratio of the infarcted area (pale) to the risk area (deep red). Distance between white columns = 1 cm. (**B**) TUNEL assay (TUNEL-positive cells, pink; DAPI, blue) was performed 1 week following MI. (**C**) Masson’s trichrome staining of cardiac sections 1 week after establishing MI (collagen fibers, light blue; muscle fibers, red). (**D**) Immunofluorescence staining for CD31 1 week after MI (CD31-positive microvessels, red; DAPI, blue). (**E**) Ejection fraction. (**F**) Representative pressure–volume loop. (**G**) End-systolic pressure–volume relationship (ESPVR). (**H**) A schematic summary of the regulation of EndMT by treating EchA in ischemic hearts. * *p* < 0.05 vs. MI group. Data are expressed as X-fold induction compared to normal control. All values are mean ± standard deviation. Statistical significance was assessed by one-way ANOVA.

## Data Availability

Data will be made available upon request.

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
