# Peer review of "Regulation of Inflammation-Mediated Endothelial to Mesenchymal Transition with Echinochrome a for Improving Myocardial Dysfunction"

_marinedrugs, 2022, doi:10.3390/md20120756_

Round 1

Reviewer 1 Report

The manuscript by Song and colleagues demonstrates with in vitro (HUVEC) and in vivo (MI) models the anti-fibrotic effect of EchA induced by the combination of 2 proinflammatory cytokines, TGF-b2 and IL1b

The manuscript is well organized, and the data are well organized and convincing

Minor concerns

Why the choice of using the combination of two cytokines to induce EndMT ? This should be reported in the introduction

Figure 5C should be better described and the quality improved. The panels are small and by increasing the size they lose resolution. With such quality it is impossible to appreciate the different expression of VE cadherin and alpha-SMA in the MI model toward the MI model treated with EchA.

As above, Figure 6D. The image is very dark, and this referee has difficulty believing that there is a difference in microvascular density between the two panels.

Last aspect: the in vivo model, both the molecular and functional analysis part is little discussed, when in fact it is an important demonstration of the antifibrotic efficacy in the cardiovascular context of the product the authors describe. I suggest discussing the results better.

Author Response

  1. Why the choice of using the combination of two cytokines to induce EndMT ? This should be reported in the introduction.

Ans) Thank you for your comments. It is important to mention for this study, so we mentioned in chapter 2.1. of Result section, however not in Introduction section. We added this issue [lane 57-59].

  1. Figure 5C should be better described and the quality improved. The panels are small and by increasing the size they lose resolution. With such quality it is impossible to appreciate the different expression of VE cadherin and alpha-SMA in the MI model toward the MI model treated with EchA.

Ans) We strongly agree with the reviewer about the need for good image data. We tried to improve the quality of the images and corrected Figure 5C accordingly.

  1. As above, Figure 6D. The image is very dark, and this referee has difficulty believing that there is a difference in microvascular density between the two panels.

Ans) We strongly agree with the reviewer about the need for good image data. We tried to improve the control of image contrast and corrected Figure 6D accordingly.

  1. Last aspect: the in vivo model, both the molecular and functional analysis part is little discussed, when in fact it is an important demonstration of the antifibrotic efficacy in the cardiovascular context of the product the authors describe. I suggest discussing the results better.

Ans) Thank you for your kind suggestions. It is very important to analyze antifibrotic efficacy based on molecular and functional analysis in in vivo level. After examining the changes in the MI area according to EchA treatment in the Results and Discussion section, we tried to confirm and discuss the final data showing antifibrotic function. Following the reviewer's suggestion, we added a follow-up discussion to solidify the in vivo results [lane 328-330 and 348-353].

Reviewer 2 Report

This is an excellent study and will undoubtedly generate new knowledge and could be potential therapeutic strategy. However, authors need to address some critical points below.

1.       I cannot see any data that treated with IL-1B alone

2.       Is EchA treatment increased IL-10 and IL-13 production?

3.       Regarding mitochondria, it is essential however to show the status of mitochondrial DNA content and autophagy markers (LC3II, LC3I) following EchA treatment.

4.       In figure 5, what is the status of periostin level?

5.       In figure 6 , cardiac function data Ejection fraction is not so convincing. What about the status of GLS and FS? What is the status of the a-SMA expression post MI following EchA treatment?

Author Response

  1. I cannot see any data that treated with IL-1B alone

Ans> We conducted a study based on the EndMT induction phenomenon according to the co-regulation of TGF-β2 and IL-1β in ischemic cells (Ref. 4). Therefore, please understand that all processing was carried out under the condition of co-stimulation.

  1. Is EchA treatment increased IL-10 and IL-13 production?

Ans> Thank you for your opinion. This manuscript is the result of studying the suppression of myocardial fibrosis by controlling EndMT response by EchA. Changes in inflammatory response can be predicted according to changes in interleukin proteins, but it was not confirmed because it is not a major response. However, we think there is a possibility enough because a study that induces anti-inflammatory response by treating EchA in skin cells has been reported (Mar Drugs. 2021 Nov 1;19(11):622). We will conduct in an additional study.

  1. Regarding mitochondria, it is essential however to show the status of mitochondrial DNA content and autophagy markers (LC3II, LC3I) following EchA treatment.

Ans> Thank you for your suggestion. It is very important issue for functional assessment of mitochondrial dysfunction. Mitochondria can regulate autophagy through the generation of ROS (Physiology (Bethesda). 2008 Oct;23:248-62). To properly study mitochondrial function, it is important to identify autophagy regulation, namely mitophagy, in this study, TMRE and mitochondrial biogenesis-regulated genes were identified in terms of weakening "function" according to mitochondrial ROS. We ask for your deeply understanding.

  1. In figure 5, what is the status of periostin level?

Ans> Thanks for outing the important point. It is known that myofibroblasts express some unique ECM proteins such as periostin as well as ECM remodeling enzymes such as matrix metalloproteases (MMPs), tissue inhibitors of metalloproteinases (TIMPs), and lysyl oxidase (LOX) (Circulation. 2012 Apr 10;125(14):1795-808. Front Physiol. 2020 May 6;11:416). All points were examined, but the periostin level was not confirmed. We will confirm the aspect of these changes through further research.

  1. In figure 6 , cardiac function data Ejection fraction is not so convincing. What about the status of GLS and FS? What is the status of the a-SMA expression post MI following EchA treatment?

Ans> We totally agree with the reviewer's opinion. However, we are unlikely to be able to respond to the reviewer's comments. Instead of using echocardiography, we measured cardiac function using millar catheterization, which can confirm microdynamic sensing between blood pressure and blood flow. That is why we cannot present GLS and FS. However, additional data, 1) dP/dt (a signal derived from pressure, typically ventricular blood pressure that indicates the change in pressure over time) and 2) Ves (End-systolic Volume) also proved that our results are meaningful response to the effects of EchA. And, the status of a-SMA is represented in Figure 5C.

Figure. Data of millar catheterization. *P<0.01, **P<0.05.

Reviewer 3 Report

I read with great interest the paper “Regulation of Inflammation-Mediated Endothelial to Mesenchymal Transition with Echinochrome A for Improving Myocardial Dysfunction”.

Paper design is fine. The article is logically divided into sections and subsections. Figures are of good quality. English needs to be revised. Data presented are of good interest.

Comments:

1.      Introduction: a better background should be provided on the role of inflammation in myocardial dysfunction development as it is a fundamental part of the paper. You can find some information about inflammation and heart remodelling in several settings in these two reviews (doi: 10.31083/J.RCM2305165; doi: 10.3390/biom11121834)

2.       it should be reported the strength and limitation of this study.

Author Response

  1. Introduction: a better background should be provided on the role of inflammation in myocardial dysfunction development as it is a fundamental part of the paper. You can find some information about inflammation and heart remodelling in several settings in these two reviews (doi: 10.31083/J.RCM2305165; doi: 10.3390/biom11121834)

Ans> Thank you for the reviewer’s suggestions. It is very important issue of the role of inflammation in myocardial dysfunction. So, we added their citation with some information [lane 66-68; Ref. 11 and 12].

  1. it should be reported the strength and limitation of this study.

Ans> The strengths of this study are written in the conclusion part at the end of the discussion. However, we did not write any limitations in conducting this study. In accordance with the reviewer's opinion, some limitations have been described together [lane 372-373].

Round 2

Reviewer 2 Report

No further concern. Thanks for addressing my points